# Perceptions of the seriousness of major public health problems during the COVID-19 pandemic in seven middle-income countries

Richard T. Carson[1], Michael Hanemann[2], Gunnar Köhlin [3✉], Wiktor Adamowicz [4], Thomas Sterner[3], Franklin Amuakwa-Mensah [3], Francisco Alpizar[5], Emily A. Khossravi[1], Marc Jeuland [6], Jorge A. Bonilla [7], Jie-Sheng Tan-Soo[8], Pham Khanh Nam [9], Simon Wagura Ndiritu[10], Shivani Wadehra[11,16], Martin Julius Chegere[12], Martine Visser[13], Nnaemeka Andegbe Chukwuone [14] & Dale Whittington [15✉]

## Abstract

**Introduction** Public perception of the seriousness of the COVID-19 pandemic compared to six other major public health problems (alcoholism and drug use, HIV/AIDS, malaria, tuberculosis, lung cancer and respiratory diseases caused by air pollution and smoking, and water-borne diseases like diarrhea) is unclear. We designed a survey to examine this issue using YouGov's internet panels in seven middle-income countries in Africa, Asia, and Latin America in early 2022.

**Methods** Respondents rank ordered the seriousness of the seven health problems using a repeated best-worst question format. Rank-ordered logit models allow comparisons within and across countries and assessment of covariates.

**Results** In six of the seven countries, respondents perceived other respiratory illnesses to be a more serious problem than COVID-19. Only in Vietnam was COVID-19 ranked above other respiratory illnesses. Alcoholism and drug use was ranked the second most serious problem in the African countries. HIV/AIDS ranked relatively high in all countries. Covariates, particularly a COVID-19 knowledge scale, explained differences within countries; statistics about the pandemic were highly correlated with differences in COVID-19's perceived seriousness.

**Conclusions** People in the seven middle-income countries perceived COVID-19 to be serious (on par with HIV/AIDS) but not as serious as other respiratory illnesses. In the African countries, respondents perceived alcoholism and drug use as more serious than COVID-19. Our survey-based approach can be used to quickly understand how the threat of a newly emergent disease, like COVID-19, fits into the larger context of public perceptions of the seriousness of health problems.

## Plain Language Summary

We were curious what people in different countries thought about the seriousness of COVID-19 compared to other health problems. We designed a survey, and hired YouGov, a survey research firm, to administer it in seven countries in Africa, Asia, and Latin America in early 2022. Respondents answered the questions on their computer, tablets, or smart phones. Their answers revealed that in most countries respiratory illnesses were perceived to be a more serious problem than COVID-19. In Africa people felt that alcoholism and drug use were also more serious than COVID-19. These findings are important because they show that people still care about the health problems they were facing before the pandemic, which is useful information for healthcare providers.

[1] University of California, San Diego, La Jolla, CA, USA. [2] Arizona State University, Tempe, AZ, USA. [3] University of Gothenburg, Gothenburg, Sweden. [4] University of Alberta, Edmonton, Alberta, Canada. [5] Wageningen University and Research, Wageningen, Netherlands. [6] Duke University, Durham, NC, USA. [7] Universidad de los Andes, Bogotá, Colombia. [8] National University of Singapore, Singapore, Singapore. [9] University of Economics, Ho Chi Minh City, Vietnam. [10] Strathmore University, Nairobi, Kenya. [11] Ashoka University, Noida, India. [12] University of Dar es Salaam, Dar es Salaam, Tanzania. [13] University of Cape Town, Cape Town, South Africa. [14] University of Nigeria Nsukka, Nsukka, Nigeria. [15] University of North Carolina at Chapel Hill, Chapel Hill, NC, USA. [16] Present address: UPES, Dehradun, India. ✉email: gunnar.kohlin@efd.gu.se; Dale_Whittington@unc.edu

The global public health community has undertaken periodic efforts to compare health problems along different dimensions and to assess the effectiveness of various health interventions. Most existing health prioritization exercises have been expert assessments of the seriousness of different health problems or the attractiveness of alternative health interventions using perspectives such as cost-effectiveness[1–4]. In contrast, eliciting the general public's perspectives is less common, and usually involves a single country or comparisons across high-income countries[3,5–9]. There have been few cross-country comparisons of how people in low- and middle-income countries rank the seriousness of the health problems they face[10]. Exceptions include the Kaiser-Pew Global Health Survey (conducted before the COVID-19 pandemic and not explicitly including respiratory illnesses) and a 2012 study examining public preferences for efficiency versus equity criteria conducted in Brazil, Cuba, Nepal, Norway and Uganda[11,12]. The available evidence suggests people in low- and middle-income countries often have very different assessments of the seriousness of different health problems than experts think they should have[13].

Health policymakers increasingly recognize the value of incorporating individuals' assessments about healthcare priorities. As a result, new frameworks have been proposed for incorporating the public's perception of disease risks in policymaking[14]. Understanding public assessments for disease risk is important because 1) health resource allocation decisions aligned with public assessments are more likely to receive support and be easier to implement; 2) health interventions guided by public opinion increase people's quality of life and provide "peace of mind"; and 3) people may have better information about the risks they face in their specific local context and personal realities than experts working with aggregate data[15].

In January-February 2022, we examined how people in two upper-middle-income countries (Colombia and South Africa) and five lower-middle-income countries (India, Kenya, Nigeria, Tanzania, and Vietnam) ranked the perceived seriousness of seven health problems (alcoholism and drugs; HIV/AIDS; malaria; tuberculosis; respiratory illness like lung cancer caused by air pollution and smoking; water-borne diseases like diarrhea; and the new entrant COVID-19). There is, of course, a long list of health problems in any country and there is some element of arbitrariness with any subset. Our choice here is motivated by several factors.

First, we wanted respondents to rank order the complete set provided. (See Supplementary Methods: Elicitation Approach for Ranking Severity - Repeated Best-Worst Format for discussion of issues involved in examining a larger set of problems.) Second, we intentionally excluded cardiovascular diseases and cancer. These two groups are the two largest sources of mortality in many countries and manifest themselves in many forms that also have many names, making them hard to explain adequately in a short survey question. Cardiovascular diseases and other forms of cancer also overlap with some of our other problems (e.g., respiratory diseases due to air pollution and smoking). Further, although they represent major problems for many age groups, they are widely perceived as unavoidable and problems of the elderly. We also wanted a set of problems that the public believed could be largely addressed with public health interventions. This ruled out some

major sources of mortality and morbidity, such as crime and traffic accidents.

The aim of the research was to determine whether respondents' ranking of public health issues was dominated by COVID-19, which was affecting populations in the countries surveyed and had received extensive media attention. The main result is that in every country except Vietnam, respiratory illnesses were perceived to be a more serious problem than COVID-19. Also, in the four African counties (Nigeria, Kenya, Tanzania, and South Africa), respondents reported that alcoholism and drug use was a more serious health problem than COVID-19.

## Methods

During the COVID-19 pandemic, large surveys using in-person interviews have been practically impossible to implement because respondents understandably would not allow enumerators into their homes. As a result, most survey data have been collected through internet panels. The results reported in this paper come from web surveys that YouGov implemented in Colombia, India, Kenya, Nigeria, South Africa, Tanzania, and Vietnam. These countries represent about 25% of the world's population and about 29% of the total population of low- and middle-income countries.

In each country, YouGov randomly selected 1200 members from their existing internet panels. Each sample represented that country's internet-connected population above the age of 18 years (including access through computers, mobile phones, and tablets). Respondents were interviewed through an online survey that included incentives for participation. As Table 1 shows, the internet-connected population varies across countries, and different data sources report slight differences in the percent of a country's population connected. The January 2022 Report of *Digital* aggregates information from a more diverse set of sources and generally suggests higher internet penetration in low- and middle-income countries than the World Bank[16,17]. Our results are not representative of a country's overall population. The internet-connected population tends to be somewhat younger, slightly more male than female, more educated, with higher income, and less likely to live in rural areas. However, this population is likely to be quite important in implementing health communication strategies. While our analysis should produce unbiased estimates for this sampling frame, those estimates will not be representative of the population in each country that is not connected to the internet. (Sample design and survey execution details are described in Supplementary Methods: Sample Selection, Execution, and Weighting.)

The internet survey questionnaire included a ranking exercise implemented using a best-worst question elicitation format[18]. From a list of seven health problems, respondents were asked to select the most serious and least serious (see Supplementary Methods: Elicitation Approach for Ranking Severity - Repeated Best-Worst Format, and Supplementary Methods: Initial Best-Worst Question).

These two health problems were then removed from the list and the choice task was repeated using the five remaining problems. The respondent was then asked a third time to indicate the most and least serious problems from the remaining three

| Table 1 Internet-connected percent of population by country | | | | | | | |
|---|---|---|---|---|---|---|---|
| Source | Colombia | India | Kenya | Nigeria | South Africa | Tanzania | Vietnam |
| Digital: January Report 2022 | 69% | 47% | 42% | 51% | 68% | 25% | 73% |
| World BankNovember 2022 | 70% | 43% | 30% | 36% | 70% | 22% | 70% |

problems. The respondent's answers to these three best-worst tasks provide their ranking of the seven health problems.

We used a rank-ordered logit model to analyze the data generated. The model provides one of the most widely used ways of averaging respondents' rankings. For example, assume malaria is ranked the second most serious problem by almost all respondents. However, respondents are almost equally split between the other diseases that they rank as the most serious health problem, and thus none of the health problems is ranked most serious by a majority of respondents. In this case malaria, in a manner similar to ranked-choice voting, will be predicted by a rank-ordered logit model to be the most serious disease problem. (See Supplementary Methods: Rank-Ordered Logit Model).

Much of our interest focused on how COVID-19 ranks among a well-defined set of competing health problems, so it is useful to look at measures of the severity of the pandemic across those countries. As the pandemic unfolded, there were many issues involving the reporting of COVID-19 statistics and those are beyond the scope of this paper. However, it is important to recognize that those issues varied across countries and were more prevalent in countries where the health reporting was often inadequate[19]. Perhaps just as important for our purpose here, it is not clear which of the commonly reported COVID-19 statistics societal actors were focused on. Did the number of cases or the number of deaths drive perceptions of COVID-19's threat in these countries? Table 2 presents cases and deaths expressed in terms of both absolute values and per million individuals. Kenya, Nigeria, and Tanzania reported substantially fewer COVID-19 cases and deaths than the other four countries on both a total and per capita basis. There are several explanations for this[20]. The large role played by age in COVID-19 mortality is now widely accepted and these African countries have lower fractions of their population 65 years of age and over (the most vulnerable age cohort). In the United States low temperatures have been causally related to higher number of COVID-19 cases and deaths using high-frequency reporting date-corrected data[21]. Kenya, Nigeria, and Tanzania are closer to the equator than most of our countries and hence have warmer minimum temperatures.

Moreover, both reported cases and deaths, in absolute and per capita terms, only capture part of the picture as they ignore the quantity of testing deployed relative to the magnitude of pandemic. A measure that captures both is the case fatality rate (CFR, deaths/cases). The lower the CFR, the better the testing regime. This can be seen by noting that the CFR generally converges toward the unobserved infection fatality rate (IFR) rate from above. The IFR has been identified as being well below 1% in a few ideal reporting contexts in high-income countries. Work in low- and middle-income countries suggests that the presence of fewer elderly, as a fraction of the population, offsets lower survival likelihoods, so we expect that the IFR in all of our seven countries should be well below 1%[22, 23]. If a strong testing regime is seen by the public as indicative of the severity of the problem, Vietnam with a CFR of 1.13 ranks highest.

The survey received Institutional Review Board (IRB) approval from the Swedish Ethical Review Authority, Research Ethics Board at the University of Alberta, and Research Review Board at Wageningen University in the Netherlands. Respondents gave YouGov their informed consent to participate in the survey.

**Reporting summary**. Further information on research design is available in the Nature Portfolio Reporting Summary linked to this article.

## Results

Figure 1a-g displays the predicted severity ranking of the seven health problems in each of the seven countries (data for Fig. 1 available in Supplementary Data 1). The most striking result is that the category of other respiratory diseases was the top-ranked health problem in every country except Vietnam. In Vietnam, respiratory diseases was ranked the second most serious health problem, after COVID-19. We expected that COVID-19 would be the respondents' top concern due to availability bias, i.e., that information about COVID-19 was widely available in the media and a topic of discussion in most households at the time of the survey[24]. However, this was not the case. Instead, COVID-19 occupied an effective three-way tie for the second-highest ranking health problem, along with alcohol/drugs and HIV/AIDS (see Supplementary Table 1).

Particularly striking is the second-place ranking of alcohol and drugs in the four African countries. This contrasts with their substantially lower ranking in the three non-African countries. COVID-19 and HIV/AIDs rise because they are ranked near the middle in most countries, with the first place ranking of COVID-19 in Vietnam pushing COVID-19 up and keeping alcohol and drugs from a clear overall second place ranking.

Respondents perceived tuberculosis, malaria, and water-borne diseases as less serious threats. This may be partly explained by the fact that our sample frame includes a larger fraction of urban, higher income, and more educated individuals than each country's overall population. Thus, water-borne illness may not be as important a problem in our internet-connected population because poorer, rural households are underrepresented.

Figure 2 displays the results for perceived seriousness from a rank-ordered model (see Supplementary Methods: Rank-Ordered Logit Model and Supplementary Data 2). Interestingly, the average of the ranks of the two reported COVID-19 statistics (cases per capita and CFR) has a correlation of 0.95 ($p < 0.01$) with the seriousness ranking. The rank ordering of the March 1, 2022 vaccine rates (at least one dose) reported by Johns Hopkins University's Coronavirus Resource Center [Colombia (79.5%), India (68.2), Kenya (14.4), Nigeria (8.1), South Africa (34.1), Tanzania (4.9), Vietnam (80.5)] has a correlation of 0.93 ($p < 0.01$) with our seriousness ranking.

| Table 2 COVID-19 statistics* | | | | | |
|---|---|---|---|---|---|
| Country | Cumulative Reported Cases | Cases Per Million | Cumulative Deaths | Deaths Per Million | Case Fatality Rate (%) |
| Colombia | 6,065,801 | 116,933.29 | 138,854 | 2,676.75 | 2.29 |
| India | 42,938,599 | 30,298.77 | 514,246 | 362.87 | 1.20 |
| Kenya | 322,978 | 5978.03 | 5639 | 104.37 | 1.75 |
| Nigeria | 254,570 | 1164.86 | 3142 | 14.38 | 2.50 |
| South Africa | 3,675,691 | 61,370.06 | 99,430 | 1660.10 | 2.70 |
| Tanzania | 33,620 | 513.30 | 798 | 12.18 | 2.37 |
| Vietnam | 3,557,629 | 36,233.25 | 40,338 | 410.83 | 1.13 |

*As of March 1, 2022, taken from https://ourworldindata.org/explorers/coronavirus-data-explorer, whose source is the Johns Hopkins University Coronavirus Resource Center.

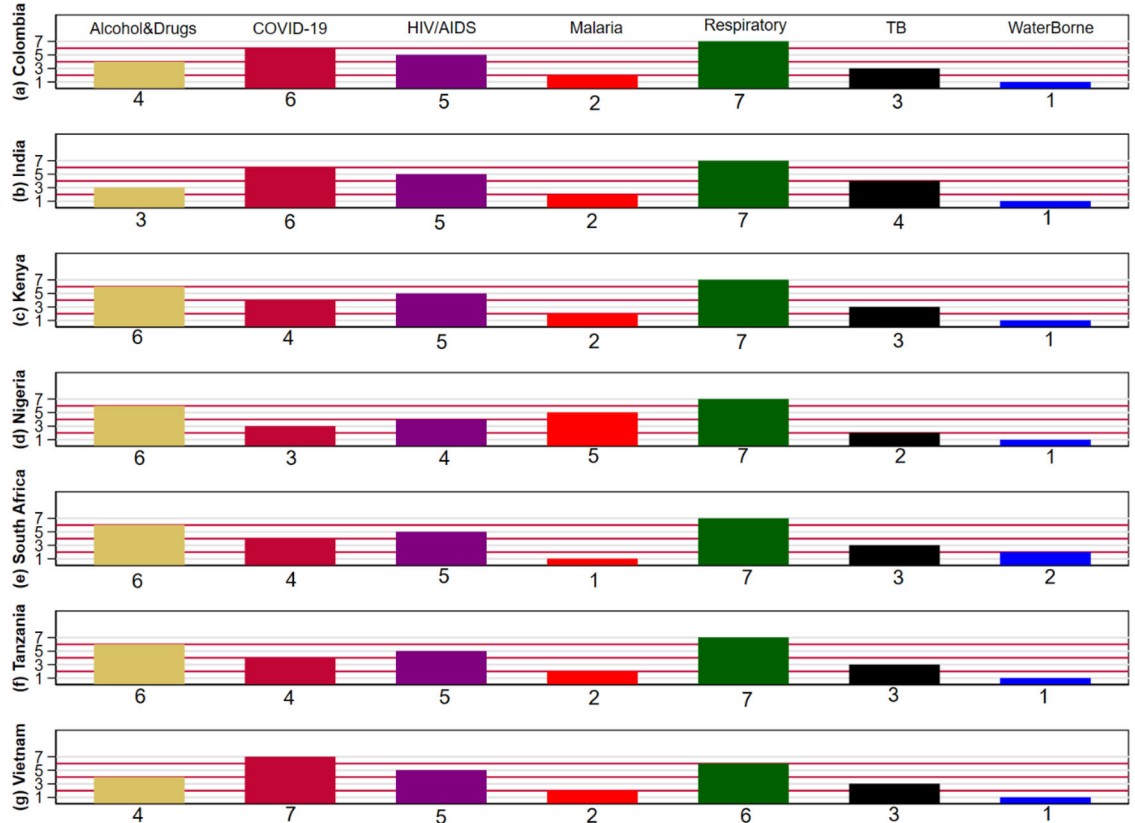

**Fig. 1 Seriousness of health problems in your community over the next five years (Colombia, India, Kenya, Nigeria, South Africa, Tanzania, and Vietnam).** Each panel (**a–g**) in this figure shows the average relative ranking of seven health problems by respondents in one of the seven countries based on the rank-ordered logit model [**a**) Colombia, **b**) India, **c**) Kenya, **d**) Nigeria, **e**) South Africa, **f**) Tanzania, **g**) Vietnam.] The rank of 7 represents the health problem ranked most serious, while the rank of 1 represents the least serious health problem.

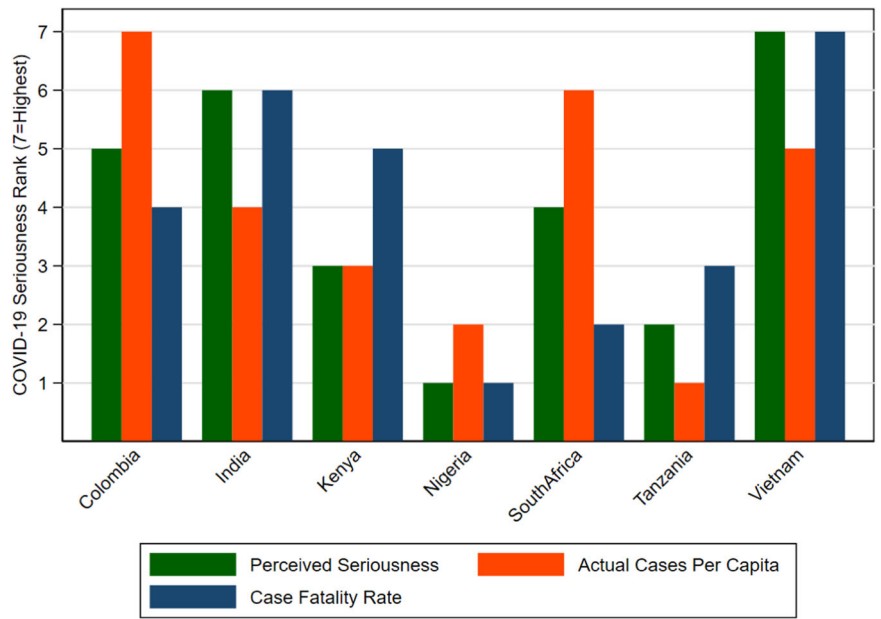

**Fig. 2 Actual measures versus public perception of seriousness of COVID-19 (Colombia, India, Kenya, Nigeria, South Africa, Tanzania, and Vietnam).** Bars reflect the rank ordering across countries of the perceived seriousness score from our model, seriousness as reflected in per capita COVID-19 cases measured by testing, and the case fatality rate. The lowest case fatality rate is given the highest rank (7th), where this measure reflects seriousness from a public health perspective through deploying a high testing rate relative to cases and deaths. Cumulative COVID-19 cases & deaths as of 1 March 2022 from Johns Hopkins Coronavirus Research Center.

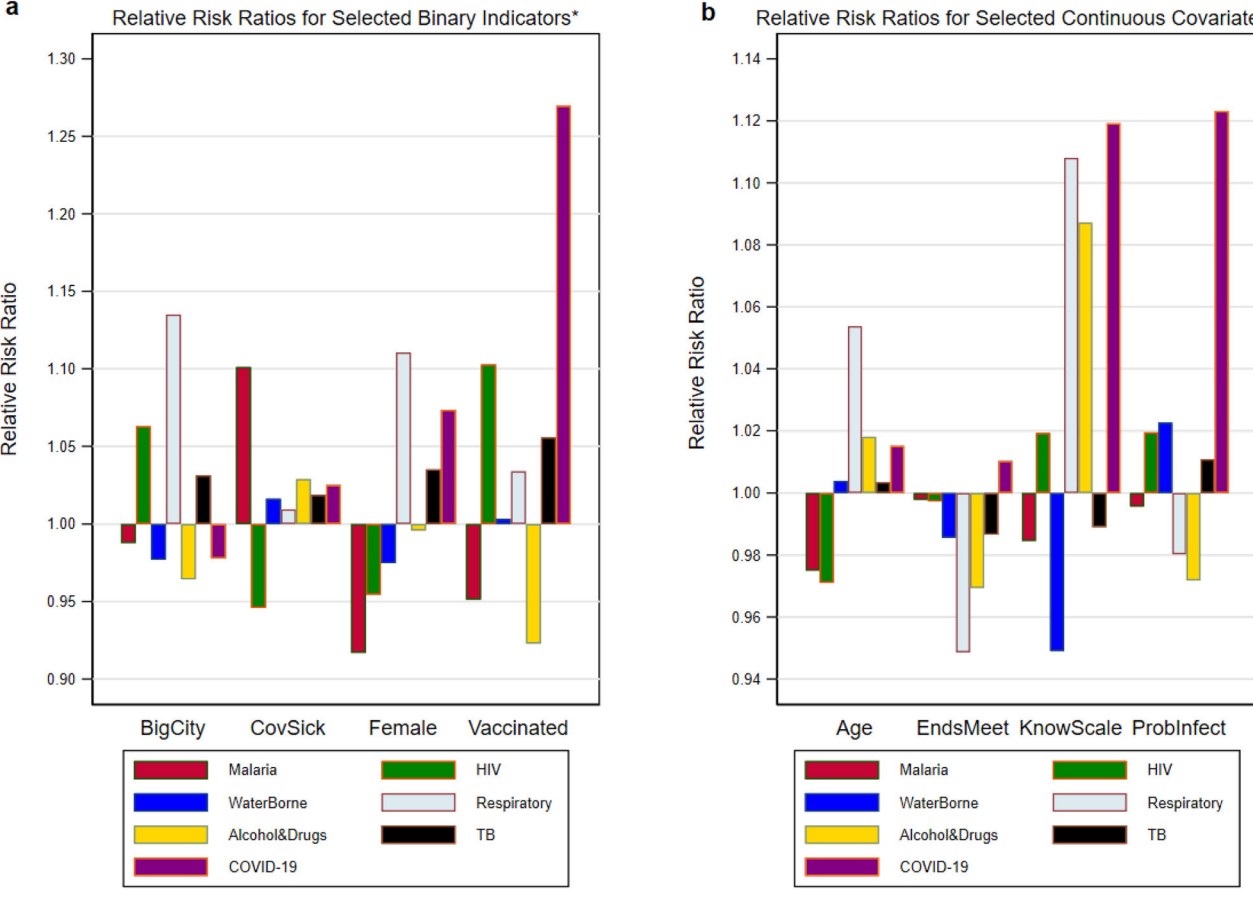

**Fig. 3 Covariate risk ratios for individual health problems.** Relative risk ratios for respondent covariates are derived from the rank-ordered logit model with country-fixed effects and respondent covariates. In Panel **a** displayed effect sizes represent a binary shift from 0 to 1 in the indicator variable. In Panel **b** displayed effect size estimates are based on a one standard deviation change in the value of the covariate x model coefficient.

To examine the relationship between respondents' perceptions of the seriousness of health problems/diseases and their socio-economic and demographic characteristics, we re-estimate (see Supplementary Methods: Rank-Ordered Logit Model) the rank-ordered logit model adding respondent covariates[9]. Figure 3a presents relative risk parameters (1 = no difference) for four binary variables (gender, living in a large city, having been infected with COVID-19, and having been vaccinated); Fig. 3b presents relative risk parameters for four continuous and categorical variables (age; difficulty of making ends meet; a scale measuring respondent's knowledge of COVID-19; and respondent's perceived probability of being infected with COVID-19 in the future).

Interpreting the relative ratios (the exponentiated coefficients from the rank order logit model) is different for binary and continuous variables. For the binary variables, the coefficient represents the shift from the "0" level (e.g., male) to the "1" level (e.g., female). For the continuous variables an alternative notion is needed, and we employ the typical one standard deviation measure (see Supplementary Methods).

The most striking finding in Panels (a) and (b) of Fig. 3 is that many of these relative risk ratios are of modest size once overall country-specific differences are taken into account. The larger relative risk ratios are all statistically significant at conventional levels, but the smaller ones are not (Supplementary Methods). Looking at the specific binary indicators, respondents in large cities rank respiratory diseases higher than respondents living elsewhere. Women rank respiratory diseases and COVID-19 higher than men. Having been vaccinated has almost no influence

on rankings. Intriguingly, respondents' rankings of malaria and respiratory diseases are higher if someone in the respondent's household has had COVID-19.

A one standard deviation change in age or a household's difficulty of making ends meet had little influence on health problem rankings. However, this was not true of the two COVID-19 variables. Those who were more knowledgeable about COVID-19 rank both COVID-19 and respiratory diseases higher. However, the largest effect on the COVID-19 ranking was for the perceived likelihood of getting infected with COVID-19 in the next 12 months. For this COVID-19 variable, a one standard deviation increase was associated with a sizeable increase in respondents' ranking of the seriousness of the COVID-19 health problem. This finding would not have been apparent by examining past COVID-19 infection or vaccination status.

## Discussion

The COVID-19 pandemic has had profound direct and indirect effects on healthcare systems throughout the world. The demand for COVID-19 care has crowded out care for other diseases. At the same time, in many countries the supply of healthcare services has decreased, as healthcare personnel have suffered from COVID-19 and a substantial increase in workloads. In addition to these supply and demand pressures on healthcare systems, there has been an enormous media focus on the COVID-19 epidemic. Thus, it would not have surprised us to find respondents pre-occupied with COVID-19, to the neglect of other health problems.

Results from this multi-country study of disease and the seriousness of health problems show that respondents in seven middle-income countries are quite concerned about COVID-19. In all seven countries respondents ranked COVID-19 as more serious than traditional health concerns like malaria, tuberculosis, and water-borne illnesses. However, the emergence of this new infectious disease and all the associated media attention has still not placed COVID-19 at the top of most respondents' rankings. That distinction belongs to other respiratory diseases associated with air pollution and smoking, a finding with implications for how health authorities should think about addressing respiratory diseases. It also clearly suggests that the COVID-19 pandemic has not completely crowded out concern about other serious health problems. To the contrary, knowledge of COVID-19 increases concern about other respiratory diseases. In our four African countries, alcohol and drugs stand out as a health problem of particular concern.

The newness of COVID-19 helps illustrate that pandemic statistics influence public perceptions. Vietnam is an instructive example, where COVID-19 is ranked the most serious of the seven health problems. Yet Vietnam has experienced fewer cases or deaths per capita than Colombia and South Africa (Table 2), and likely fewer than in India where deaths appear to have been substantially under reported[25]. This may be due to the Vietnamese health authorities' early aggressive actions requiring masking and testing and their continued vigilance. This is reflected in Vietnam's very low CFR. The higher correlation between the country-level perceived seriousness ranking and the average of the two objective rankings (number of cases per capita and case fatality ranking), than either objective measure alone, suggests that perceived seriousness was influenced by multiple objective COVID-19 statistics. This insight may be useful to efforts that try to characterize how health systems respond to respiratory pandemics[26].

In our survey, respondents were asked how they assessed the seriousness of different health problems, not how they thought health sector resources should be allocated to address these health problems. Nevertheless, we believe that our results are important information for health policy decision-makers to consider when allocating resources. Our results show that respondents have different assessments of the relative importance of health problems than some experts believe. While the whole world focused almost exclusively on COVID-19, many respondents saw other respiratory problems and alcohol and drugs as the most important health problems.

An important lesson for health bureaucracies is to not get too carried away by what media sources report at a particular point in time. It is important to avoid crowding out ordinary health services. The larger point is that public perceptions of the seriousness of health problems can be multifaceted with considerable heterogeneity within and across countries and population segments defined by demographics and knowledge. Public perceptions of the severity of health problems are only one consideration in the formulation of health policy. It should never be dispositive. Obviously, other factors should be taken into account. These include the need to consider how effective additional resources are in reducing a particular health problem and how different interventions affect the equity and distribution of health outcomes. When divergences occur between public perceptions of the seriousness of a health problem and the budget priorities of health authorities, transparent communication by health authorities can be helpful in maintaining public support.

A potential weakness of our work is the choice of sampling frame. Our results are only representative of the internet-connected population in each of our seven countries. It is worth noting that such a sampling frame is likely to come to dominate future survey work in low- and middle-income countries, just as it already has in high-income countries, due to cost and completion time considerations. Issues related to in-person surveys during a pandemic have only added to the forces pushing survey work in this direction. This internet-connected population is also likely the main target of many health communication efforts. As is the case in high-income countries, understanding the ways in which the internet-connected part of the population differs from the rest of the population is now an important task for the survey research community. One of the primary strengths of this survey mode is that it is much more replicable from both a cost and time perspective. Hence, it would be possible to alter or expand the set of health problems examined in our research. Perhaps the most interesting application is a straightforward replication of our best-worst question format of health problems to understand how public perceptions of the seriousness of COVID-19 are changing as the pandemic moves toward an endemic situation.

## Data availability

The numerical data underlying Figs. 1–3 is contained in Supplementary Data 1. The data sets analyzed during the current study are available from the Environment for Development (EfD), Univeristy of Gothenburg website: https://snd.gu.se/en/catalogue/collection/efd.

## Code availability

The codes used for the analyses are also available from the Environment for Development, University of Gothenburg website: https://snd.gu.se/en/catalogue/collection/efd.

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

## Acknowledgements
We thank Alexander Marsolais, YouGov, for his advice on the design and implementation of the survey. Funding for this study was provided by the Swedish International Development Cooperation Agency (SIDA) through the Environment for Development Initiaitve (EfD) at the University of Gothenburg.

## Author contributions
D.W., R.C., and M.H. conceived of the experiment; D.W., R.C., M.H., G.K., W.A., T.S., F.A., F.A.M., M.J., J.T., and E.K. designed the survey instrument; J.B., P.N., S.N., S.W., M.C., M.V., and N.C. adapted the survey instrument to country conditions; R.C. carried out the data analysis; and D.W. and R.C. co-wrote the paper.

## Funding

## Competing interests
The authors declare no competing interests.
