## [Peer Review File · Communications Medicine]

Reviewers' comments:

Reviewer #1 (Remarks to the Author):

This paper aims at describing the prioritization of health issues in 7 developing Countries in Africa, Asia, and Latin America during the pandemic. The investigation was performed in 2022 through a web-based survey. The comparative analysis involving 7 developing countries is interesting. However, there are some significant considerations to do.

1) The authors didn't report the public health contexts (the Covid-19 pandemic evolution and the widespread of the other infections) in the 7 Countries they compared. This lacking description sounds as an important limitation of the study according to its "main objective to determine whether respondents' rankings would be dominated by a newly emergent infectious disease (COVID-19) that was affecting populations in the countries surveyed and that was receiving extensive media attention". An accurate description of the public health context (including the prevalence of the different health problems) would improve the interpretation of the survey result.

2) The authors included alcohol and drugs among the investigated health problems. What is the reason considering that the other are infective diseases (HIV/AIDS, malaria, TB, other respiratory illness, water-borne diseases, and COVID-19)?

3) The demographic characteristics and the vaccination status of the respondent populations from the 7 countries should be better detailed at least as supplementary material.

4) At line 272 and followings the authors should consider also the evidence form the literature reported by F. Cascini, I. Hoxhaj, D. Zaçe, M. Ferranti, M.L. Di Pietro, S. Boccia, W. Ricciardi. How health systems approached respiratory viral pandemics over time: a systematic review. *BMJ global health* 2020, 5 (12), e003677

5) About the considerations on political interventions inspired by the results of this survey, the authors should better explain this point taking into account the points 1) and 4) above.

Reviewer #2 (Remarks to the Author):

Brief summary of the manuscript

This paper presents the results of an online survey of respondents from 7 low- and middle-income countries in which respondents were asked to rank the relative priority of a selection of diseases using a best-worst ranking methodology. The findings show that respondents perceive non-COVID respiratory illnesses to be the most serious problem facing their country in all countries except Viet Nam, where respiratory illnesses were second to COVID.

Overall impression of the work

Overall, this is a thoughtful, well-organized work that contributes something new to the literature. It presents an interesting use of an existing data gathering resource that has clear potential to be applied to similar questions of prioritization by citizens in LMICs. The questions addressed are timely and valuable. The statistical analyses look appropriate and I have no concerns about the authors

ability to reproduce the work in light of the level of detail provided in the main document and online materials.

The two major areas of concern that I would recommend the authors address in the manuscript is the focus on COVID and placing the sample frame and sample population in context.

1. While I understand the desire to highlight the results seemingly most relevant to current events, the focus in the results section on COVID and its comparison to respiratory illnesses obscures the relative importance placed on other conditions, namely alcohol and drugs and HIV/AIDS. This is particularly evident in the 4 African countries. This could be rectified by expanding the discussion to include why alcohol and drugs and HIV/AIDS have such a prominent role, discussing the differences between country-specific findings more clearly, or reducing the frequency of references to COVID.

2. The main context the authors offer is a comment on the age and urban lean of their sample frame. They state that the typical internet connected population skews younger and male. (I note that the median age of the sample is 31.9, (29.4-33.5) but given that the median age in Nigeria is ~18 years old, and ~70% of the country is under 30 the sample doesn't actually appear to be much younger than would be expected). They also mention that the sample includes more urban and higher income and education respondents. Given how small the eligible population was in most countries as a fraction of the general population (particularly in India, Kenya, Nigeria, and Tanzania), more detail is warranted on the magnitude of the differences between the general population and the populations from which the sample was sourced. Sharing how much more urban, educated, wealthy the respondents were compared to their country peers would help the reader better understand and place the findings in context. (Hopefully this wouldn't be too burdensome as it appears most of the sample-specific data is readily available from the covariate analysis.)

Specific comments, with recommendations for addressing each comment

The following are minor comments offered for consideration to make the paper stronger.

- The sentence "Most existing health prioritization exercises have been expert assessments of the seriousness of different health problems [1] or the attractiveness of different health interventions [2]." starts by claiming that "most" prioritization exercises are of a limited type, but offers only 2 examples. Given the substantial range of methods that are used to assess "the attractiveness of different health interventions" additional citations would help.

o <https://www.cambridge.org/core/journals/international-journal-of-technology-assessment-in-health-care/article/criteria-used-for-prioritysetting-for-public-health-resource-allocation-in-low-and-middleincome-countries-a-systematic-review/76F388E6474C7FF51F8682BD5961FD1D>

o <https://www.tandfonline.com/doi/full/10.1080/23288604.2015.1123338>

- Figure 1 would be easier to interpret if the relative ranking was added in/below/near each of the bars in the graph. It is hard to qualitatively assess the heights of bars that are situated far away from each other.

- The choice of these seven diseases, rather than cancer, heart disease, and other chronic conditions, would benefit from more explanation. Currently the justification is because these other diseases are most common among elderly, and less likely to prompt immediate new public health interventions, but prior to this there is no mention of age as a criterion or point of focus for the paper. These are significant areas to exclude given that cardiovascular disease and neoplasms are 2 of the top 3 leading causes of death among 15-49-year-olds in India, Viet Nam, and Colombia (the other being injuries).

- "YouGov randomly selected 1200 members from their existing internet panels." A link or reference where the reader could learn more about the construction of these panels would be helpful.

- Malaria and TB figures: Perhaps I'm not understanding these correctly but the text "Figure 2 displays the relative rankings of reported per capita prevalence and the severity of malaria across the seven countries based on coefficients from the best-worst model" seems to suggest that the green bars would be equivalent to the relative ranking of the disease in the best-worst exercise, but, for example, in Nigeria malaria appears to be ranked 3rd according to figure 1 but scores a 7 in Figure 2 panel a.

- Some references seem to be missing from the online materials (specifically with respect to the data on the internet connected population per country).

Reviewer #3 (Remarks to the Author):

Authors developed BWS survey to Prioritize Health Issues During the COVID-19 Pandemic in Seven Developing Countries. It is an interesting topic, but the clarity of the manuscript need to be improved. I have some comments and questions for authors' considerations.

1-Please mention all abbreviations in the abstract and manuscript.

2-It is better to use a structured format for paper. Currently, the different parts (Introduction, Methods, results, ...) are not separated and is not clear.

3-There are different types of BWS method including:

BWS object case

BWS Profile case

BWS Multi-profile case

in the method, mention which type was used and explain the reasons.

4-Regarding the selected choice tasks, explain what's your reference. Why did not use a balanced choice tasks?

5-I recommend using the table for BWS results and ranking.

6-Similar to Figure 1, the ranking of health problems can be shown according to the results of all countries.

Reviewer #4 (Remarks to the Author):

The paper presents the results of a multi country web survey of lay people's views on the severity of seven health problems. The seven countries were low to mid income countries in Asia and Sub-Saharan Africa, and the seven considered health problems were alcoholism and drug use, malaria, respiratory diseases, TB, covid-19, HIV/aids, and water borne diseases.

How people in a country rank the severity of diseases/health problems provides useful information, for reasons the authors present in the introduction (support for policy, risk assessments and more), hence, knowledge about this is welcome. The authors correctly claim that there exists rather few studies of this, yet, I think they underestimate what is actually done in this field. There are several studies for high income countries, and some from low income countries. None of these are referred to. (I list a few below for illustration - there are more).

The title of the paper is "How people prioritize ...". I believe the respondents were asked to rank order health problems according to severity, not how they would prioritize between them. This might be considered the same, but this is (not always) the case. A reasonable interpretation of prioritization is that it involves policy, e.g. the allocation of resources between measures to address these health problems.

The methodology of the paper is well described, and adequate. In fact, I was surprised by the relatively high response rates in Columbia, SA and Vietnam. The authors discuss the biases, especially in terms of age and wealth, yet, the variations in the response rates and the skewed samples within each country, deserves more attention.

This is highly relevant for the implications of the findings. I would like to mention two aspect of this. First: If the results are clearly biased, how, if at all, can they be used in further work, be it research or policy? Secondly: Although I agree with the claim that people's views on severity are important for the reasons you mention, there are much more to this. A fundamental question is this: If the general public and the health authorities disagree on severity, whose views should constitute the basis for policy? (Why?) Further, as mentioned above, severity is one thing, priority setting of measures (allocation of resources) is another. If the general public considers, say, respiratory disease the most severe, does it automatically follow that those health problems should be prioritized?

My final comment concerns the choice of health problems. Generally, I think it was a good idea to concentrate on diseases that are "less likely to prompt immediate new public health interventions", yet I am less sure that all of those you chose are likely to prompt new interventions. Also, respiratory diseases is the only one that includes cancer, which I believe may have added to its top rank position. Despite these comments, I think a revised paper deserves to be published. I hope my comments can contribute to an improved version.

Some studies of experts and lay people's views on priorities:

Solberg, C.T., Tranvåg, E.J. & Magelssen, M. Attitudes towards priority setting in the Norwegian health care system: a general population survey. *BMC Health Serv Res* 22, 444 (2022).

<https://doi.org/10.1186/s12913-022-07806-9>

Winkelhage, J. , Schreier, M. and Diederich, A. (2013) Priority setting in health care: Attitudes of physicians and patients. *Health*, 5, 712-719. doi: 10.4236/health.2013.54094.

Sabik LM, Lie RK. Priority setting in health care: Lessons from the experiences of eight countries. *Int J Equity Health*. 2008 Jan 21;7:4. doi: 10.1186/1475-9276-7-4. PMID: 18208617; PMCID: PMC2248188.

Peacock, S.J. Public attitudes and values in priority setting. *Isr J Health Policy Res* 4, 29 (2015). <https://doi.org/10.1186/s13584-015-0025-8>

Kapiriri L, Norheim OF. Criteria for priority-setting in health care in Uganda: exploration of stakeholders' values. *Bull World Health Organ*. 2004 Mar;82(3):172-9. Epub 2004 Apr 16. PMID: 15112005; PMCID: PMC2585925.

Kapiriri, L., Martin, D.K. Priority setting in developing countries health care institutions: the case of a Ugandan hospital. *BMC Health Serv Res* 6, 127 (2006). <https://doi.org/10.1186/1472-6963-6-127>

Response to Reviewers' comments

Reviewer #1

This paper aims at describing the prioritization of health issues in 7 developing Countries in Africa, Asia, and Latin America during the pandemic. The investigation was performed in 2022 through a web-based survey. The comparative analysis involving 7 developing countries is interesting. However, there are some significant considerations to do.

Comment 1 - The authors didn't report the public health contexts (the Covid-19 pandemic evolution and the widespread of the other infections) in the 7 Countries they compared. This lacking description sounds as an important limitation of the study according to its "main objective to determine whether respondents' rankings would be dominated by a newly emergent infectious disease (COVID-19) that was affecting populations in the countries surveyed and that was receiving extensive media attention". An accurate description of the public health context (including the prevalence of the different health problems) would improve the interpretation of the survey result.

Authors' response: We thank Reviewer #1 for this suggestion. In the revised manuscript, we discuss the COVID situation in each of the countries at the time of the survey. Data on both number of COVID-19 cases and deaths are now provided so that the reader can better interpret our results. We have also added a new table (Table 1) to address Reviewer No. 1's comment. The new Table 2 in the revised manuscript provides a standard set of statistics which is then used for the construction of a new Figure 2.

Comment 2 - The authors included alcohol and drugs among the investigated health problems. What is the reason considering that the other are infective diseases (HIV/AIDS, malaria, TB, other respiratory illness, water-borne diseases, and COVID-19)?

Authors' response: Our intent here was to provide a standard set of current health problems as a reference point for respondents to consider and not to limit this set to infectious diseases. Our respiratory disease category is described as due to air pollution and smoking, and is also not an infectious disease category. The exact nature of the set of health problems is not critical for our purpose, which was to see if COVID-19 crowded out a set of other health problems. In the revised text we explain why we did not want to include cancer and cardiovascular disease because both are too broad and overlap with our respiratory disease category which explicitly references lung cancer as an exemplar. We did not want to include heart disease because while it is important for many age groups, it is concentrated in the elderly. We did not want to include various types of accidents, including motor vehicles, because a reduction in accidents is often not perceived as a typical public health intervention.

Comment 3 - The demographic characteristics and the vaccination status of the respondent populations from the 7 countries should be better detailed at least as supplementary material.

Authors' response: We now provide this information for India drawing on the large full probability sample done for the India in the 2019 Pew's Global Value Survey. These data were used for weighting purposes by YouGov since they include internet access status as well as a set of demographic variables. India falls at the lower end of the fraction of the population with internet access. We explain that official statistics available from international organizations like the United Nations and World Bank are not useful for this purpose since they are based on either the entire population (and that our sample as well as almost all

passing standard Institutional Review Board approvals) is restricted to those age 18 and over. Thus, relevant statistics are conditional on that restriction. Alternatively, age brackets which span the age of 18 [lower end of the relevant bracket is 15 and upper end is above 18] are used. Information on the fraction of the population receiving at least one vaccine dose in each of the countries as of 1 March 2022 is now included in the revised manuscript.

Comment 4- At line 272 and followings the authors should consider also the evidence form the literature reported by F. Cascini, I. Hoxhaj, D. Zaçe, M. Ferranti, M.L. Di Pietro, S. Boccia, W. Ricciardi. How health systems approached respiratory viral pandemics over time: a systematic review. *BMJ global health* 2020, 5 (12), e003677

Authors' response: In the revised manuscript we note in the conclusion that the tension between COVID-19 statistics related to prevalence and prevention are likely to be useful to analyses like Cascini, et al. (2020) of how health systems respond to respiratory pandemics, and we have cited Cascini et al (2020).

Comment 5- About the considerations on political interventions inspired by the results of this survey, the authors should better explain this point taking into account the points 1) and 4) above.

We thank Reviewer #1 for this suggestion and have revised the Discussion section to address this issue.

Reviewer #2:

The two major areas of concern that I would recommend the authors address in the manuscript is the focus on COVID and placing the sample frame and sample population in context.

Comment 1. While I understand the desire to highlight the results seemingly most relevant to current events, the focus in the results section on COVID and its comparison to respiratory illnesses obscures the relative importance placed on other conditions, namely alcohol and drugs and HIV/AIDS. This is particularly evident in the 4 African countries. This could be rectified by expanding the discussion to include why alcohol and drugs and HIV/AIDS have such a prominent role, discussing the differences between country-specific findings more clearly, or reducing the frequency of references to COVID.

Authors' response: The revised paper has been expanded and now includes a much more substantial consideration of the findings beyond COVID and respiratory diseases due to air pollution and smoking. In particular, the striking finding of concern about alcoholism and drug use in the African countries is emphasized as recommended by Reviewer #2, as is the broad concern across countries for HIV/AIDS.

Comment 2. The main context the authors offer is a comment on the age and urban lean of their sample frame. They state that the typical internet connected population skews younger and male. (I note that the median age of the sample is 31.9, (29.4-33.5) but given that the median age in Nigeria is ~18 years old, and ~70% of the country is under 30 the sample doesn't actually appear to be much younger than would be expected). They also mention that the sample includes more urban and higher income and education respondents. Given how small the eligible population was in most countries as a fraction of the general population (particularly in India, Kenya, Nigeria, and Tanzania), more detail is warranted on the magnitude of the differences between the general population and the populations from which the sample was

sourced. Sharing how much more urban, educated, wealthy the respondents were compared to their country peers would help the reader better understand and place the findings in context. (Hopefully this wouldn't be too burdensome as it appears most of the sample-specific data is readily available from the covariate analysis.)

Authors' response: The specific median age issue raised here is due to the fact that typically reported median ages for a country are for the entire population. Our sample has the typical Institutional Review Board restriction on surveys that respondents have to be 18 years or older. As such our median age is the median age of those 18 and over.

In the main body of the text, we note that current estimates put the fraction of the population with internet access well above the somewhat dated figures YouGov provides us with which are taken from the survey sources used to construct weights. The comparisons suggested are somewhat difficult because they cannot be based on official government statistics such as those reported to the United Nations or World Bank. Those are typically based on the entire population rather than those 18 and over. When more detailed information by age is available, the lower end of age bracket containing the relevant population starts at 15 and not 18 and has a higher upper end than 18. The relevant information could be extracted from YouGov's original sources for weighting but most are not publicly available. We have access to the 2019 Pew Global Value Survey for India which was YouGov's source for weighting information for that country through an academic affiliation. India falls toward the lower end of the fraction of the population with internet access among our 7 LMIC countries. In the Online Materials we now provide a detailed discussion of how those with and without internet access differ on age and gender as well as on a measure of income and urbanicity.

Specific comments, with recommendations for addressing each comment

The following are minor comments offered for consideration to make the paper stronger.

- The sentence "Most existing health prioritization exercises have been expert assessments of the seriousness of different health problems [1] or the attractiveness of different health interventions [2]." starts by claiming that "most" prioritization exercises are of a limited type, but offers only 2 examples. Given the substantial range of methods that are used to assess "the attractiveness of different health interventions" additional citations would help.

o <https://www.cambridge.org/core/journals/international-journal-of-technology-assessment-in-health-care/article/criteria-used-for-prioritysetting-for-public-health-resource-allocation-in-low-and-middleincome-countries-a-systematic-review/76F388E6474C7FF51F8682BD5961FD1D>

o <https://www.tandfonline.com/doi/full/10.1080/23288604.2015.1123338>

Authors' response: Introduction has been revised to make distinctions clear and these citations are now included in the paper.

- Figure 1 would be easier to interpret if the relative ranking was added in/below/near each of the bars in the graph. It is hard to qualitatively assess the heights of bars that are situated far away from each other.

Authors' response: Figure 1 has been revised to include these.

- The choice of these seven diseases, rather than cancer, heart disease, and other chronic conditions, would benefit from more explanation. Currently the justification is because these other diseases are most common among elderly, and less likely to prompt immediate new public health interventions, but prior to this there is no mention of age as a criterion or point of focus for the paper. These are significant areas to exclude given that cardiovascular disease and neoplasms are 2 of the top 3 leading causes of death among 15–49-year-olds in India, Viet Nam, and Colombia (the other being injuries).

Authors' response: In the introduction we now provide an expanded discussion of our choice of the six other health problems for inclusion in the comparison set with COVID-19. We make it clear that the purpose is not to provide a complete ranking ordering of a very large set of health problems, but rather was to examine whether the extensive attention being paid to COVID-19 caused it to dominate an easy-to-understand set of competitors. (A description of some of the technical issues involved in doing this is now included in the Online Materials.) The rationale for why cardiovascular diseases and cancer have been excluded has been expanded in the revised manuscript.

- “YouGov randomly selected 1200 members from their existing internet panels.” A link or reference where the reader could learn more about the construction of these panels would be helpful.

Authors' response: In the Online Material's we now provide a link to YouGov's response to ESOMAR.org (the relevant professional organization) standardized set of 28 questions that major survey firms fill out that provide extensive details on this and related issues.

- Malaria and TB figures: Perhaps I'm not understanding these correctly but the text “Figure 2 displays the relative rankings of reported per capita prevalence and the severity of malaria across the seven countries based on coefficients from the best-worst model” seems to suggest that the green bars would be equivalent to the relative ranking of the disease in the best-worst exercise, but, for example, in Nigeria malaria appears to be ranked 3rd according to figure 1 but scores a 7 in Figure 2 panel a.

Authors' response: We have dropped the Malaria and TB figures in favor of providing a similar figure for COVID-19 that provides greater continuity with the rest of the paper. We have also included a more extensive discussion of how to interpret the figure. The main source of confusion here was that in Figure 1, the ranking is across the seven health problems within each country, while in Figure 2, the continuous score from the rank-ordered logit model is used to allow comparisons across countries. The Online Materials also contains further discussion of this issue.

- Some references seem to be missing from the online materials (specifically with respect to the data on the internet connected population per country).

Authors' response: There are now three sets of numbers concerning the fraction of the population that is connected to the internet. In the initial version of the main paper, we provide a set of estimates from the World Bank and a reference to their document. We also provide another set of estimates from one of standard industry sources for all seven of our countries which suggests that the internet penetration rate for the three central African countries is substantially higher than that reported to the World Bank. In the Online Material's we now make clear that the “internet penetration rate” is that which is provided by YouGov (so there is no reference for this), and that this rate is somewhat out of date, having been derived from various full-probability survey samples undertaken between 2017 and 2019.

Reviewer #3

Comment 1: Authors developed BWS survey to Prioritize Health Issues During the COVID-19 Pandemic in Seven

Comment 1- Please mention all abbreviations in the abstract and manuscript.

Authors' response: This has been done for BWS and LMIC. We have eliminated the use of TB in the manuscript and instead refer to "tuberculosis" everywhere.

Comment 2 - It is better to use a structured format for paper. Currently, the different parts (Introduction, Methods, results, ...) are not separated and is not clear.

Authors' response: We have made these changes in the revised manuscript as requested by Reviewer #3

Comment 3- There are different types of BWS method including:

BWS object case

BWS Profile case

BWS Multi-profile case

in the method, mention which type was used and explain the reasons.

Authors' response: We have explained in the revised manuscript is that our application of BWS is Type 1, i.e., BWS object case. In the revised manuscript we provide a brief explanation that health problems are the objects and that there are no attributes.

Comment 4- Regarding the selected choice tasks, explain what's your reference. Why did not use a balanced choice tasks?

Authors' response: No experimental design was used since respondents each receive an initial BWS task plus two follow-ups, which provides a complete ranking of the seven health problems for each respondent. A discussion of how an experimental design could be used to rank order a larger set of health problems is now provided in the Online Material.

Comment 5- I recommend using the table for BWS results and ranking.

Authors' response: As requested by Reviewer #3, we have provided this table in the revised manuscript for the first BWS task (Table 2 in the Online Materials). We also provide more explanation of why this simple display format tends to break down when combining a sequence of BWS tasks that result in a complete ranking and that the standard approach for dealing with this issue is to use a rank-ordered logit model.

Comment 6- Similar to Figure 1, the ranking of health problems can be shown according to the results of all countries.

Authors' response: It is definitely possible to stack the individual country samples. This is done for the rank-order logit models as reported in the Online Materials. Instead of estimating the first reported model allowing each country to have its own set of parameters for each of the seven health problems, a model with only a single set of alternative specific constants for those problems can be estimated. This is now included in a footnote in the Online Materials and its parameter estimates describe a well-defined ordering that characterize our sample as a whole. That rank ordering is: respiratory illness, COVID, alcohol/drugs, HIV/AIDS, TB, malaria, water-borne. That model is clearly rejected using a likelihood ratio test in favor of the country-specific version used in the paper, however. The simpler version might still be of interest if our set of countries were representative of some well-defined aggregate (e.g., countries were all chosen at random from Africa proportionate to population), but that is not the case in our research.

Reviewer #4

Comment 1 - The authors correctly claim that there exists rather few studies of this, yet, I think they underestimate what is actually done in this field. There are several studies for high income countries, and some from low income countries. None of these are referred to. (I list a few below for illustration - there are more). ...

Some studies of experts and lay people's views on priorities:

Solberg, C.T., Tranvåg, E.J. & Magelssen, M. Attitudes towards priority setting in the Norwegian health care system: a general population survey. BMC Health Serv Res 22, 444 (2022). <https://doi.org/10.1186/s12913-022-07806-9>

Winkelhage, J. , Schreier, M. and Diederich, A. (2013) Priority setting in health care: Attitudes of physicians and patients. Health, 5, 712-719. doi: 10.4236/health.2013.54094.

Sabik LM, Lie RK. Priority setting in health care: Lessons from the experiences of eight countries. Int J Equity Health. 2008 Jan 21;7:4. doi: 10.1186/1475-9276-7-4. PMID: 18208617; PMCID: PMC2248188.

Peacock, S.J. Public attitudes and values in priority setting. Isr J Health Policy Res 4, 29 (2015). <https://doi.org/10.1186/s13584-015-0025-8>

Kapiriri L, Norheim OF. Criteria for priority-setting in health care in Uganda: exploration of stakeholders' values. Bull World Health Organ. 2004 Mar;82(3):172-9. Epub 2004 Apr 16. PMID: 15112005; PMCID: PMC2585925.

Kapiriri, L., Martin, D.K. Priority setting in developing countries health care institutions: the case of a Ugandan hospital. BMC Health Serv Res 6, 127 (2006). <https://doi.org/10.1186/1472-6963-6-127>

Authors' response: We appreciate Reviewer #4's guidance on this literature. The introduction of the revised manuscript paper now contains a more extensive discussion of the issues here and now cites these papers suggested by Reviewer #4.

Comment 2 - The title of the paper is " How people prioritize ...". I believe the respondents were asked to rank order health problems according to severity, not how they would prioritize between them. This might be considered the same, but this is (not always) the case. A reasonable interpretation of

prioritization is that it involves policy, e.g. the allocation of resources between measures to address these health problems.

Authors' response: On reflection, we agree with this comment by Reviewer #4. In response to this comment, we have changed the paper title to: "How People Perceive the Seriousness of Major Public Health Issues During the COVID-19 Pandemic: Evidence from Seven Developing Countries"

Comment 3 - The methodology of the paper is well described, and adequate. In fact, I was surprised by the relatively high response rates in Colombia, SA and Vietnam. The authors discuss the biases, especially in terms of age and wealth, yet, the variations in the response rates and the skewed samples within each country, deserves more attention.

This is highly relevant for the implications of the findings. I would like to mention two aspect of this. First: If the results are clearly biased, how, if at all, can they be used in further work, be it research or policy?

Authors' response: We thank Reviewer #4 for this comment. In the revised manuscript, we now include a much more extensive discussion of survey-related issues. There are several major issues. The first is that for standard research purposes surveys employing high-quality, full-probability random samples of the adult population in most developed and developing countries are now prohibitively expensive. Survey researchers thus define sampling frames from which they can draw random samples. Surveys based on such samples can provide unbiased estimates of the responses of those selected to be interviewed. They are by construction not representative of the fraction of the general population that is not included in the sampling frame. As such we clearly note that the paper results are not biased for the sampling frame of the internet-connected population in each country and that this part of the population is of substantive interest from the perspective of health communication strategies in its own right. The issue here is best cast in terms of how representative the non-internet connected population is.

Comment 4: Although I agree with the claim that people's views on severity are important for the reasons you mention, there are much more to this. A fundamental question is this: If the general public and the health authorities disagree on severity, whose views should constitute the basis for policy? (Why?) Further, as mentioned above, severity is one thing, priority setting of measures (allocation of resources) is another. If the general public considers, say, respiratory disease the most severe, does it automatically follow that those health problems should be prioritized?

Authors' response: We now provide more discussion of this issue in the conclusion emphasizing that the public's perspectives should be one of the inputs to decision making by health authorities and that there are other considerations ranging from how effective additional resources are at reducing a health problem to equity.

Comment 5 - My final comment concerns the choice of health problems. Generally, I think it was a good idea to concentrate on diseases that are "less likely to prompt immediate new public health interventions", yet I am less sure that all of those you chose are likely to prompt new interventions. Also, respiratory diseases is the only one that includes cancer, which I believe may have added to its top rank position.

Authors' response: There is now an extensive discussion on the selection of the health problems for the comparison set for COVID-19. The use of lung cancer as an exemplar for respiratory diseases is specifically mentioned there. In the revised manuscript we have dropped the phrase "less likely to promote immediate new public health interventions" because we agree with Reviewer #4 that it is potentially confusing and not specifically referenced in any of the question wording respondents saw.

Reviewers' comments:

Reviewer #1 (Remarks to the Author):

The authors clarified the points requested through the review process and improved the manuscript accordingly. I would recommend the publication of the paper.

Reviewer #2 (Remarks to the Author):

The authors have done a thoughtful and thorough job of responding to the peer reviews (mine and others), and should be commended for their work. I find the revised manuscript to be clear and their findings well-supported by the analyses they present. While I don't agree with their characterization of several of the omitted diseases (e.g., that prevention of road traffic accidents is not perceived as a public health issue), I think their justifications for the choice of conditions is defensible. I also appreciate the attention given to the unexpectedly high ranking of alcohol and drug use in the African countries. Despite the limitations in generalizability, which the authors appropriately acknowledge, I think this paper will provide an interesting addition to the literature base in the priority-setting field.

Though I know we were not asked to provide comments on minor errors, I could not help but track a few while re-reviewing. I offer them below to help facilitate the author's finalization efforts should the paper be accepted.

- Methods, paragraph 5 "Did the number of cases or the number of deaths drive perceptions of COVID-19's threat in these counties?" The final word should be countries
- Methods, paragraph 6, "(CFR, cases/deaths)" - should be deaths/cases
- For Figure 1a-g, I find the numbering a bit counterintuitive. I'd suggest adding a figure note explaining that 1=Least severe to 7=Most severe
- Figure 1g (Vietnam), I believe the number under the green bar (Respiratory) should be 6, not 7

[EDITORS NOTE: This reviewer indicated all previous points raised by Reviewer 3 except points 4 and 5 have been adequately addressed.]

Reviewer #4 (Remarks to the Author):

Despite improvements after the reviewers' comments, I am still hesitant to recommend publication. I have three reasons for this: First, the study did not ask for how to prioritize resources, only the respondents' views on the relative severity of diseases. This is now reflected in a new title, yet not in the discussion. Secondly, the jump from severity ranking to priority setting reflects a shallow understanding of the relationship between the two. Finally, the discussion of how public views should be taken into account does not relate to the discussion of this topic in the literature.

[ED NOTE: I discussed these comments with reviewer 2, who does not think these concerns are as serious an issue. This reviewer has suggested you rephrase the 3 most problematic sentences in the Discussion:

"Our results show that respondents have different priorities for health policy interventions than some experts believe. While the whole world focused almost exclusively on COVID, many respondents saw respiratory problems and alcohol and drugs as the most important health problems." And "Public perceptions of which health issues should be prioritized and receive more resources, is but one consideration in the formulation of health policy."

This reviewer recommends you lightly rephrase to avoid the implication that participants' responses to this exercise are a direct reflection of their priorities for health policy in their countries."]

Reviewer #5 (Remarks to the Author):

1. While reviewer 3 (Point 3) points out that there are different types of best-worst scaling, only one type is appropriate for a prioritization task like this. Several recent reviews have noted that in this case, best-worst scaling or BWS used in the generic is the most used term to describe this method. I believe that focusing on other types of BWS or choice experiment is not appropriate here and that using terms like case 1 and object case that are not commonly used in the literature does not help readers of this type of work. BWS does appear to be appropriate here and if one was to ask why it was chosen, it should be in the context of other methods that specifically focus on prioritization.
2. Point 4 by review 3 is an important one. BWS is a choice experiment, and as such it should have an appropriate experimental design. Even in the abstract, there is a signal that something out of the norm is being done here. Specifically, it states that "rank ordered the seriousness of the seven health problems using a repeated best- worst scaling (BWS) method". Two concerns for me are flagged here, on is that BWS is a scoring approach rather than a ranking approach. Second, as an experiment using an experimental design, it is unclear why one would describe it as a "repeated best-worst scaling (BWS) methods".
3. Upon review of the methods, this appears to be a ranking exercise, with repeated best-worst responses. Ranking is an appropriate method for prioritization, but I don't think that it should be called best-worst scaling and it is not consistent with being called a type 1 BWS. This said, the authors are not the first to call this ranking procedure best-worst scaling – see Ratcliffe, J., Kaambwa, B., Hutchinson, C. et al. Empirical Investigation of Ranking vs Best–Worst Scaling Generated Preferences for Attributes of Quality of Life: One and the Same or Differentiable?. Patient 13, 307–315 (2020).
4. In my opinion, this study would best be described as a ranking exercise with alternating best and worst choices. I think it would be advisable to use terms like "prioritization", rather than "preferences" here. I think there is growing interest in issues of prioritization, especially in public health. In this study, one could have biased the results by present attributes that varied across study team defined levels (as might have been the case in a DCE, conjoint analysis, AHP, BWS preference methods).
5. I also think that the use of a rank-ordered logit is commonly associated with ranking exercises, while BWS can use an array of other simple and advanced techniques. I assume that estimator the authors used did not control for heterogeneity within the sample. More traditionally types of mixed logits are being used to analyze studies like this – especially if there is within group heterogeneity. While the important of accounting for heterogeneity is not as important in BWS, I think some discussion (or even presentation) of other estimation methods may have been warranted.
6. Finally, I am concerned that this study focuses too heavily on ranking as the primary results. Even in a ranking exercise conducted in two or more people, one can generate a score, rather than just a simple rank. Presenting only ranking can hide important issues related to strength of effects (as maybe seen in a cardinal or ordinal scoring system). It can also prevent one from exploring scale differences across the groups (with such different potentially stemming from differences in preference heterogeneity or from differences in measurement error).

Response to Reviewers - manuscript entitled, "How People Perceive the Seriousness of Major Public Health Problems During the COVID-19 Pandemic: Evidence from Seven Developing Countries"

Reviewer #1 (Remarks to the Author):

The authors clarified the points requested through the review process and improved the manuscript accordingly. I would recommend the publication of the paper.

Authors' Response: We appreciate Reviewer #1's recommendation to publish the paper.

Reviewer #2 (Remarks to the Author):

The authors have done a thoughtful and thorough job of responding to the peer reviews (mine and others), and should be commended for their work. I find the revised manuscript to be clear and their findings well-supported by the analyses they present. While I don't agree with their characterization of several of the omitted diseases (e.g., that prevention of road traffic accidents is not perceived as a public health issue), I think their justifications for the choice of conditions is defensible. I also appreciate the attention given to the unexpectedly high ranking of alcohol and drug use in the African countries. Despite the limitations in generalizability, which the authors appropriately acknowledge, I think this paper will provide an interesting addition to the literature base in the priority-setting field.

Though I know we were not asked to provide comments on minor errors, I could not help but track a few while re-reviewing. I offer them below to help facilitate the author's finalization efforts should the paper be accepted.

- Methods, paragraph 5 "Did the number of cases or the number of deaths drive perceptions of COVID-19's threat in these counties?" The final word should be countries

Authors' response: In the revised manuscript, we have changed "counties" to "countries". We appreciate Reviewer #2's careful reading of our manuscript.

-Methods, paragraph 6, "(CFR, cases/deaths)" - should be deaths/cases

Authors' response: In the revised manuscript, we have changed "cases/deaths" to "deaths/cases".

-For Figure 1a-g, I find the numbering a bit counterintuitive. I'd suggest adding a figure note explaining that 1=Least severe to 7=Most severe

Authors' response: In the revised manuscript, to respond to Reviewer #2's request, we have added a figure note explaining that 1=Least severe to 7=Most severe

-Figure 1g (Vietnam), I believe the number under the green bar (Respiratory) should be 6, not 7

Authors' response: In the revised manuscript, we have changed the number under the green bar (Respiratory) from "7" to "6".

Reviewer #4 (Remarks to the Author):

Despite improvements after the reviewers' comments, I am still hesitant to recommend publication. I have three reasons for this: First, the study did not ask for how to prioritize resources, only the respondents' views on the relative severity of diseases. This is now reflected in a new title, yet not in the discussion.

Secondly, the jump from severity ranking to priority setting reflects a shallow understanding of the relationship between the two. Finally, the discussion of how public views should be taken into account does not relate to the discussion of this topic in the literature.

"Our results show that respondents have different priorities for health policy interventions than some experts believe. While the whole world focused almost exclusively on COVID, many respondents saw respiratory problems and alcohol and drugs as the most important health problems." And "Public perceptions of which health issues should be prioritized and receive more resources, is but one consideration in the formulation of health policy."

This reviewer recommends you lightly rephrase to avoid the implication that participants' responses to this exercise are a direct reflection of their priorities for health policy in their countries."]

Authors' response: In the revised manuscript, we have addressed Reviewer #4's concerns. We have rephrased the "most problematic" sentences in the Discussion to read as follows:

In our survey, respondents were asked how they assessed the seriousness of different health problems, not how they thought health sector resources should be allocated to address these health problems. Nevertheless, we believe that our results can be important information for health policy decisionmakers to consider when allocating resources. Our results show that

respondents have different assessments of the relative importance of health problems than some experts believe. While the entire world focused almost exclusively on COVID, many respondents saw respiratory problems and alcohol and drugs as the most important health problems in their countries.

This is an important lesson for health bureaucracies not to get too carried away by what the world media reports at a particular point in time. It is important to avoid crowding out ordinary health services. The larger point is that public perceptions of the seriousness of health problems can be multifaceted with considerable heterogeneity within and across countries and population segments defined by demographics and knowledge. Public perceptions of the severity of health problems are only one consideration in the formulation of health policy. It should never be dispositive. Obviously, other factors should be considered. These include the need to consider how effective additional resources are in reducing a particular health problem and how different interventions affect the equity and distribution of health outcomes. When divergences occur between public perceptions of the seriousness of a health problem and the budget priorities of health authorities, transparent communication by health authorities can be helpful in maintaining public support.

Reviewer #5 (Remarks to the Author):

1. While reviewer 3 (Point 3) points out that there are different types of best-worst scaling, only one type is appropriate for a prioritization task like this. Several recent reviews have noted that in this case, best-worst scaling or BWS used in the generic is the most used term to describe this method. I believe that focusing on other types of BWS or choice experiment is not appropriate here and that using terms like case 1 and object case that are not commonly used in the literature does not help readers of this type of work. BWS does appear to be appropriate here and if one was to ask why it was chosen, it should be in the context of other methods that specifically focus on prioritization.
2. Point 4 by review 3 is an important one. BWS is a choice experiment, and as such it should have an appropriate experimental design. Even in the abstract, there is a signal that something out of the norm is being done here. Specifically, it states that "rank ordered the seriousness of the seven health problems using a repeated best- worst scaling (BWS) method". Two concerns for me are flagged here, one is that BWS is a scoring approach rather than a ranking approach. Second, as an experiment using an experimental design, it is unclear why one would describe it as a "repeated best-worst scaling (BWS) methods".
3. Upon review of the methods, this appears to be a ranking exercise, with repeated best-worst responses. Ranking is an appropriate method for prioritization, but I don't think that it should be called best-worst scaling and it is not consistent with being called a type 1 BWS. This said, the authors are not the first to call this ranking procedure best-worst scaling – see Ratcliffe, J., Kaambwa, B., Hutchinson, C. et al. Empirical Investigation of Ranking vs Best–Worst Scaling Generated Preferences for Attributes of Quality of Life: One and the Same or

Differentiable?. Patient 13, 307–315 (2020).

4. In my opinion, this study would best be described as a ranking exercise with alternating best and worst choices. I think it would be advisable to use terms like “prioritization”, rather than “preferences” here. I think there is growing interest in issues of prioritization, especially in public health. In this study, one could have biased the results by present attributes that varied across study team defined levels (as might have been the case in a DCE, conjoint analysis, AHP, BWS preference methods).

Authors’ response: Points 1, 2, 3, and 4 are all in many ways the same comment, although they often reference distinct parts of our paper. The reviewer’s main point is that we collected rank-ordered data rather than the standard (partial ranking) data of a single best-worst question and that our use of the term best-worst scaling is in this context is confusing. We agree with the reviewer and have revised the manuscript to avoid this source of confusion.

In the revised manuscript (including the methods section) and in the revised online materials, we now clearly state that we conducted a ranking exercise. Furthermore, we note that the rank-ordered data on perceived severity is collected using a repeated best-worst elicitation format rather than using the term “best-worst scaling”. Reviewer #5 is correct that an elicitation format and a statistical scoring method are not the same thing. The reviewer is also correct that the two are often used interchangeably in the literature. The revised paper is now clear that main results are based on a rank-ordered logit statistical model while the best-worst scaling scores are based on the response to the first of three repetitions on progressively smaller choice sets of best-worst questions. The simplicity of the calculation of the best-worst scaling measures with partially-ranked data and ease of interpretation are the technique’s major attractions, but neither extends easily to fully ranked data, which is why we use the standard rank-ordered logit approach.

The term “experiment” no longer appears in the revised manuscript. Our original use of the term “experiment” was technically correct and is consistent with much of the literature, where assigning a randomly chosen set of units to the same full factorial experimental design (e.g., all 7 health problems) has long been seen as the limiting experimental case. However, we agree with Reviewer #5 that its use potentially causes confusion. In the online materials, we briefly discuss how one might expand the set of 7 health problems to a longer list using an experimental design. Here we specifically note the advantages of a Youden design which would provide balance in the occurrence of specific health problems across and within individual choice sets and provide a reference to a discussion of such experimental designs (Raghavarao 1988).

Again, to avoid confusion and respond to the Reviewer #5’s suggestion, we no longer use the term “preference” in referring to the data from the exercise we asked respondents to

undertake. This term is replaced with “assessment” which is a more accurate description of what was called for from respondents in our severity ranking task.

Figures 1 and 3 in the main body of the paper are based on rank-ordered logit models (whose parameter estimates are provided in the online material). BWS estimates are based on only the first of the three BW formatted questions and are presented in Table 2 in the online material.

5. I also think that the use of a rank-ordered logit is commonly associated with ranking exercises, while BWS can use an array of other simple and advanced techniques. I assume that estimator the authors used did not control for heterogeneity within the sample. More traditionally types of mixed logits are being used to analyze studies like this – especially if there is within group heterogeneity. While the important of accounting for heterogeneity is not as important in BWS, I think some discussion (or even presentation) of other estimation methods may have been warranted.

Authors’ response: Most choice data including binary and multinomial variants can be represented as completely or partially ranked data. Further, completely rank ordered data can be “exploded” into the implied binary or multinomial choice sets (Chapman and Staelin 1982). Effectively there is no information beyond the set of implied binary comparisons without making additional auxiliary assumptions. Flavors of BWS and logit models add these assumptions. In this sense, there is no difference between the two approaches, and they typically produce similar results.

In the reported work, we make the standard extreme value error component assumption that defines the standard conditional , multinomial and ranked ordered logit models. We note in the revised text that a more general model allowing for the scale parameter to vary with ranking depth, usually thought to be the main error component issue with the rank-ordered models, produces quite similar results. The robustness of models that fit aggregate share data, overall or within major divisions, has long been noted and has been formally explored in a marketing context (Allenby and Rossi, 1991). In our example, a transformation of the health problem alternative specific constants provides those market share estimates, in the sense of predicting the fraction of the population picking each problem as the most (or least) severe.

We further note in the text of the online materials that controlling for various scale parameters in the sense of a mixed or generalized multinomial logit model (Train 2009; Fiebig et al. 2010) might be useful and that such models can be fit by recognizing that rank-ordered data has an exploded logit representation which allows the fitting of such data to these model classes. The model we fit with countries and demographic covariates already has 105 estimated parameters and allowing each of these to follow a random parameter distribution (e.g., normal distributed) would double the number of estimated parameters. Allowing a full variance-covariance matrix of random components in a mixed logit model would dramatically

increase this further. This seems to us like overkill since our focus in this paper is NOT on trying to explain the fine details of the individual-level variation in the rank-ordered data.

6. Finally, I am concerned that this study focuses too heavily on ranking as the primary results. Even in a ranking exercise conducted in two or more people, one can generate a score, rather than just a simple rank. Presenting only ranking can hide important issues related to strength of effects (as maybe seen in a cardinal or ordinal scoring system). It can also prevent one from exploring scale differences across the groups (with such differences potentially stemming from differences in preference heterogeneity or from differences in measurement error).

Authors' response: Reviewer #5 raises several distinct issues raised here. The rank-ordered data cannot "hide" information on the strength of "preferences" at the individual level. That is because theoretical work in both economics and psychology has shown there is no cardinal (or interval level) information in choice data, of which a complete rank ordering is a special case.

There are several issues involved in summarizing rank-ordered data across individuals. Our paper provides results from two standard approaches: 1) a rank-ordered logit model, and 2) best-worst scaling. These are reasonably similar even though the best-worst scaling results effectively rely on a partial order [best, middle [unchosen] alternatives, worst]. There are some differences which we discuss in the paper such as the much higher fraction of the sample respondents in Vietnam ranking COVID-19 as being the most serious health problem.

We note that a third (ad hoc) approach in common usage is to average the rankings for a particular health problem, and that this also produces similar results. In our response above, we note why a model that effectively fits market shares at the country level is robust against heterogeneity in the assessments of health problems at the individual level. We also note that explicitly controlling for the most common type of measurement error difference (i.e., health problems ranked in the middle having higher level of measurement error) makes little difference suggesting that the repeated BW elicitation format we used was successful in terms of the respondents putting more effort into distinguishing between the middle alternatives. Clearly, there is another substantially different paper that could be written that delves more deeply into the nature of variability in the assessment of health problem severity by demographic covariates such as age and gender across countries, but that is outside the scope of our current paper.

Additional References Added to Online Materials Supplement

Allenby, G.M. & Rossi, P.E. There is no aggregation bias: Why macro logit models work. *Journal of Business and Economic Statistics* 9, 1-14 (1991).

Chapman, R. G., & Staelin, R. Exploiting rank ordered choice set data within the stochastic utility model. *Journal of Marketing Research*, **19**, 288-301 (1982).

Fiebig, D.G., et al. The generalized multinomial logit model: accounting for scale and coefficient heterogeneity. *Marketing Science* 29, 393-421 (2010).

Raghavarao, D. *Constructions and combinatorial problems in design of experiments* (New York: Dover, 1988).

Train, K.E. *Discrete choice methods with simulation*, 2nd ed. (New York: Cambridge University Press, 2009).

REVIEWERS' COMMENTS:

Reviewer #5 (Remarks to the Author):

The revisions have appropriately addressed the points raised by the reviewers. The revised manuscript has improved and discussion of remaining weaknesses is appropriate.